# Chirped Integrated Bragg Grating Design

**José Ángel Praena** [1,*] and **Alejandro Carballar** [2]

1   Ingeniería de Sistemas y Automática, Escuela Politécnica Superior, Universidad Pablo de Olavide, Ctra. Utrera km 1, 41013 Sevilla, Spain

2   Departamento de Ingeniería Electrónica, E.T.S. de Ingeniería, Universidad de Sevilla, C/Camino de los Descubrimientos, s/n, 41092 Sevilla, Spain; carballar@us.es

*   Correspondence: japrarod@upo.es

**Abstract:** We analyze the two classic methods for chirped Integrated Bragg Gratings (IBGs) in Silicon-on-Insulator technology using the transfer matrix method based on the effective refractive index ($n_{eff}$) technique, which translates the geometry of an IBG into a matrix of $n_{eff}$ depending on the wavelength. We also implement a procedure that allows engineering of the chirped IBG parameters, given a required bandwidth (BW) and group delay (GD). Finally, a complementary method for designing chirped IBG is proposed, showing a significant improvement in the bandwidth of the device or a moderation in the variation of the geometrical parameters of the grating.

**Keywords:** Integrated Bragg Grating; silicon photonics; chirp function; group delay; effective refractive index

## 1. Introduction

Integrated Bragg Gratings (IBGs) in Silicon-on-Insulator (SOI) technology are optical structures that implement a periodic modulation of the effective refractive index through a defined variation in the geometry of an integrated silicon waveguide [1]. They have the important advantage of being compatible with CMOS (complementary metal-oxide-semiconductor) technology manufacturing [1,2], which allows photonic and electronic circuits to be integrated into the same chip. They are very frequency-selective and have a high extinction ratio, as well as low insertion loss. IBGs are used in many photonic devices, such as sensors [3], communications [4], photonic signal processing [5], microwave photonic signal processing [6], active photonic devices, slow-light EOM (electro-optic modulators) [7,8] and dispersion control applications [9]. The latter have been especially important in recent years due to the advancement in ultrafast lasers. To achieve this purpose, apodized chirped Bragg Gratings are being developed [10].

In relation to Fiber Bragg Gratings (FBGs), chirped gratings have been widely used to address two traditional challenges: the need to broaden the bandwidth coupled by a uniform grating, and the demand for a passive device that exhibits a linear group delay as a function of wavelength that allows for chromatic dispersion compensation [11]. These two issues have been translated into IBG technology, where the design of integrated chirped gratings must take into account the wavelength dependence of the effective refractive index, as well as its geometric dependence. Compared to FBGs, these dependances constitute a setback in the design method for chirped IBGs that can be leveraged to enable new ways of designing chirped IBGs. Due to this double dependance, which can be formulated as $n_{eff}(\lambda, W)$, the traditional Coupled Mode Theory (CMT), which leaves out the physical structure of the grating and hence its variation with the effective refractive index, should not be used to analyze IBGs, as any tiny variation in the geometry has to be considered.

In this work, we propose an approach for the design of chirped IBGs based on the transfer matrix method (TMM) of electromagnetic waves in multilayer media, characterized by their effective refractive index (ERI) [12–14]. On the basis of this ERI–TMM approach,

any geometrical variation to create the IBG will be modeled, and its spectral response will be simulated. Furthermore, to know in advance the preliminary geometrical structure of the chirped IBG, an analytical and graphical procedure is presented to determine, to a first approximation, its principal geometrical parameters, relying on the $n_{eff}(\lambda,W)$ characterization of the SOI technology.

Therefore, first, we apply the ERI–TMM approach to describe and analyze the two conventional chirped IBG methods, namely, chirp via *Bragg period variation* and chirp via *waveguide width variation*. From the understanding of these two design methods, this work proposes a third new alternative to design chirped IBGs that combines the two previous methods to keep the average effective refractive index constant as a function of the coupled Bragg wavelength, $n_{eff}(\lambda_B)$, along the length of the device.

## 2. Transfer Matrix Method Based on Effective Refractive Index

Consider Figure 1a, which represents the geometrical structure of an IBG as designed using KLayout (0.26.9) software [15]. The IBG is developed on a single-mode strip-type waveguide with standard geometry: waveguide width $W$ = 500 nm (*x*-dimension, where the perturbation will be performed) and 220 nm height (*y*-dimension); the guided mode moves along the *z*-axis [1]. The parameters to be used to create the IBG, and therefore to modulate the effective refractive index along the device length, $n_{eff}(z)$, are:

1.  Corrugation shape: defines the shape of the perturbation, which can generally be rectangular or sinusoidal (as shown in Figure 1a).
2.  Corrugation width, $\Delta W$: the increase or decrease over the waveguide width (in the *x*-dimension), which represents the size of the perturbation.
3.  Bragg Period, $\Lambda_B$: length of the sidewall modulation for the geometrical perturbation (in the *z*-axis), which corresponds to the periodical variation imposed on the effective refractive index. It is worth noting that the relation between $\Lambda_B$, $n_{eff}$, and the Bragg wavelength, $\lambda_B$, is given by the Bragg condition [16]:

$$\lambda_B = 2 \cdot n_{eff}(\lambda_B, W) \cdot \Lambda_B \qquad (1)$$

4.  Grating length, *L*: together with $\Delta W$, it controls the intensity of the reflected field and the group delay for the chirped IBG.

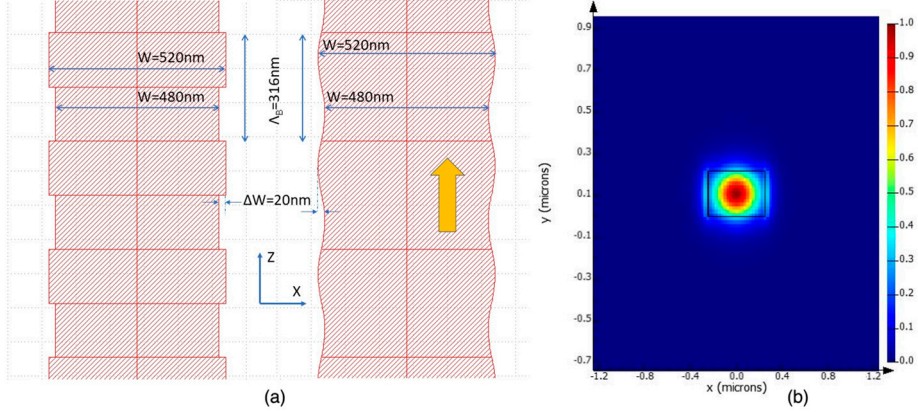

(a)  (b)

**Figure 1.** (**a**) Diagram of two Integrated Bragg Gratings (IBGs) with rectangular and sinusoidal corrugations and $\Delta W$ = 20 nm. The yellow arrow indicates the direction of propagation of the electromagnetic field. (**b**) Energy density distribution (normalized) of the quasi-transverse electric (TE) mode inside the strip waveguide (linear scale).

### 2.1. Calculation of the Effective Refractive Index as a Function of Wavelength and Waveguide Width

The effective refractive index of an integrated waveguide, $n_{eff}$, is a function of wavelength. This function can be obtained using Lumerical® (Ansys Lumerical 2021 R2.3) software [17] and depends on the waveguide geometry (in this case, strip waveguide) and the materials of which it is composed. For this computation, it is necessary to analyze the distribution of the electromagnetic field and its energy within the structure, as well as the polarization of the mode to be transmitted [1]. In the case under study, the quasi-transverse electric (TE) mode is considered, as represented in Figure 1b. Minor differences in the defined cross-sectional area can lead to small changes in the calculated $n_{eff}$. In our case, the effective refractive index obtained for $\lambda_0 = 1550$ nm is equal to 2.4525 for the geometric dimensions previously defined for the strip waveguide. After that, a sweep of wavelengths from 1500 nm to 1600 nm can be undertaken to obtain an array of $n_{eff}$ values at the considered wavelengths. From these data, a polynomial fit is performed and the following analytical expression of $n_{eff}$ as a function of $\lambda$ is obtained [1]:

$$n_{eff}(\lambda) = n_{eff}(\lambda_0) - 1,1331(\lambda - \lambda_0) - 0,035(\lambda - \lambda_0)^2 \tag{2}$$

This effective refractive index dependence on wavelength, $n_{eff}(\lambda)$, is represented in Figure 2 (light blue line). It shows a quasi-linear behavior in the spectral interval of interest, with a decrease of the $n_{eff}$ value with increasing wavelengths. The same calculation is repeated for different waveguide widths, $W$, between 450 nm and 560 nm. Figure 2 plots the effective refractive index as a function of wavelength for the different waveguide widths considered.

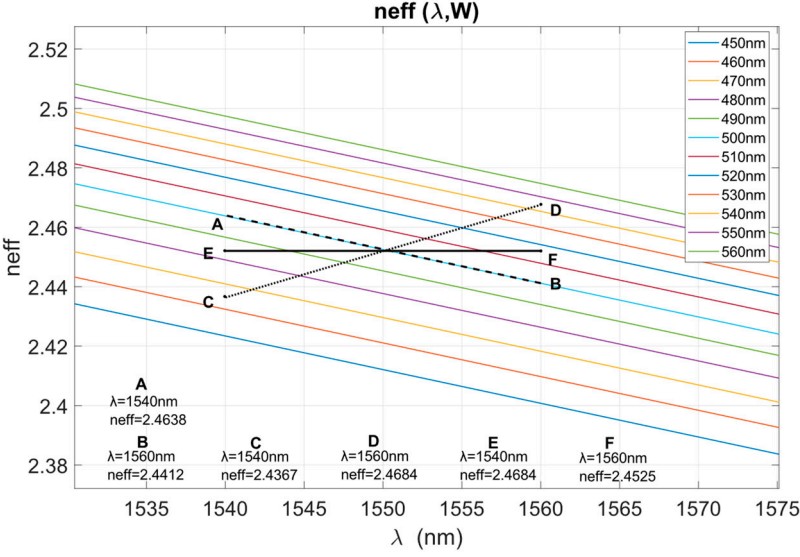

**Figure 2.** Effective refractive index obtained with numerical simulation as a function of wavelength (the discrete step wavelength is 2 nm) for different waveguide widths from 450 nm to 560 nm.

In an analogous way, Figure 3 represents the effective refractive index as a function of the waveguide width $W$, $n_{eff}(W)$, for different wavelengths. A parabolic dependence of the effective refractive index on $W$ can be observed. Both dependences of the effective refractive index, $n_{eff}(\lambda)$ represented in Figure 2 and $n_{eff}(W)$ represented in Figure 3, can be expressed as a unique two-dimensional function, $n_{eff}(\lambda,W)$, whose points can be stored in a matrix ($N$) that will be used in the modeling of IBGs by the ERI–TMM (as explained below).

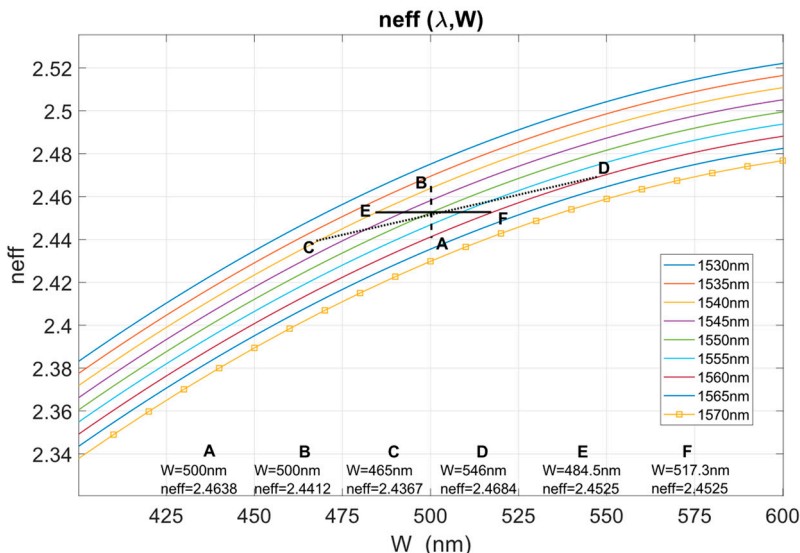

**Figure 3.** Effective refractive index obtained with numerical simulations as a function of waveguide width for different wavelengths from 1530 nm to 1570 nm (the discrete step width is 10 nm; for convenience, the sampling is marked only on the 1570 nm curve).

Figures 2 and 3 are different sides of the same coin. The linear behavior illustrated in Figure 2 is represented in Figure 3 by the constant spacing between the different wavelengths, and the quadratic variation in the $n_{eff}$ in Figure 3 is represented by the increasing spacing between the different wavelengths as the wavelength increases in Figure 2. Points A to F are used to design the chirped IBG parameters in Section 3.

### 2.2. Analysis of the IBG with the ERI–TMM

The general expression representing the geometry of the IBG in Figure 1 can be written as [14]:

$$W(z) = W_0(z) + \Delta W_{max} square\left(\frac{2\pi}{\Lambda_B}z + \varphi(z)\right) A(z) \tag{3}$$

where $W_0(z)$ is the waveguide width of the unperturbed IBG, $\Delta W_{max}$ is the maximum variation in the corrugation width, $\varphi(z)$ is the phase function that defines the phase grating chirp, and $A(z)$ is the apodization function. Both apodization and chirp functions along the grating length, $z$, are used to map the apodization and chirp profiles to the geometry of the IBG (*square* for rectangular corrugation, *sin* for sinusoidal corrugation), that is, to modulate the effective refractive index $n_{eff}(z)$ along the device length.

The process of modeling and analyzing the IBG begins with a given geometry, where $n_{eff}(z)$ must be calculated along the device length as a function of the waveguide width, $W(z)$, and considering its dependence on wavelength. To achieve this objective, the width of the waveguide as a function of $z$ is first obtained through Equation (3) and then translated into $n_{eff}(\lambda,z)$ using the previously calculated matrix $N$. The result is arranged in a new matrix, $N'$, of size $n \times m$, whose rows are the array of effective refractive index values for different positions along the IBG, $n_{eff}(z_i)$, and whose columns contain the effective refractive index dependence on wavelength, i.e., $n_{eff}(\lambda_j)$.

In that sense, $N'$ represents the function $n_{eff}(\lambda,z)$ for the IBG and characterizes the designed physical structure. The number of rows, $n$, is equal to the number of layers in which the IBG has been sampled, with the thickness of each layer $i$ ($0 < i < n$) being the differential length, $dz$, to which the transfer matrix method will be applied [13,14]. It is important to choose a sampling length that allows every detail of the physical structure to be captured, thus achieving a transfer function that is as close to real as possible. This implies selecting about 48 subdivisions for each Bragg period, which means having a $dz$ ~6 nm, corresponding to the resolution of the manufacturing process based on e-beam

lithography [14]. In this way, the IBG is divided into a number of layers (each characterized by its matrix $C_i$) and interfaces (each characterized by its matrix $I_i$) where the relationship between the incoming and outgoing fields will be governed by the following equations:

$$\begin{pmatrix} E_i^+(z_i, \lambda) \\ E_i^-(z_i, \lambda) \end{pmatrix} = I_i \begin{pmatrix} E_{i+1}^+(z_i, \lambda) \\ E_{i+1}^-(z_i, \lambda) \end{pmatrix} \tag{4}$$

$$\begin{pmatrix} E_i^+(z_{i+1}, \lambda) \\ E_i^-(z_{i+1}, \lambda) \end{pmatrix} = C_{i+1} \begin{pmatrix} E_{i+1}^+(dz, \lambda) \\ E_{i+1}^-(dz, \lambda) \end{pmatrix} \tag{5}$$

$$I_i = \frac{1}{2n_i} \begin{pmatrix} n_i + n_{i+1} & n_1 - n_{i+1} \\ n_1 - n_{i+1} & n_1 + n_{i+1} \end{pmatrix} \tag{6}$$

$$C_{i+1} = \begin{pmatrix} e^{i\lambda n_{i+1} dz} & 0 \\ 0 & e^{-i\lambda n_{i+1} dz} \end{pmatrix} \tag{7}$$

where $n_i(z, \lambda)$ and $n_{i+1}(z, \lambda)$ are the effective refractive index on both sides of each interface, and $E_i^+$ and $E_i^-$ are the incoming and outgoing fields, respectively, at the interface $i + 1$, as depicted in Figure 4b, which is a magnification of one layer of $dz$ (and the two interfaces that border it) of the entire IBG shown in Figure 4a.

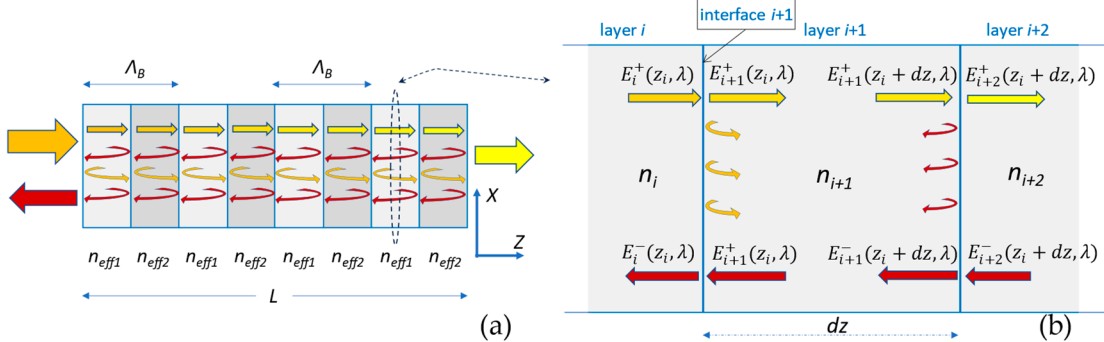

**Figure 4.** (**a**) Illustrative diagram of an IBG, showing incoming (orange), transmitted (yellow), and reflected (red) fields. (**b**) Schematic representation of the IBG sampling, and distribution of transmitted and reflected fields in a layer of size $dz$ limited by two interfaces. $L = dz \times n$ ($n$, number of layers).

Every pair interface plus layer results in a matrix $M_i$ given by the expression:

$$M_i = I_i \cdot C_{i+1} \quad 0 < i \leq n \tag{8}$$

whose coefficients are complex functions that depend on the wavelength. The matrix characterizing the behavior of the entire IBG is calculated as the product of all $M_i(\lambda)$ along the length of the device, obtaining the total matrix $M_T(\lambda)$:

$$M_i(\lambda) = \begin{pmatrix} t_{11i}(\lambda) & t_{12i}(\lambda) \\ t_{21i}(\lambda) & t_{22i}(\lambda) \end{pmatrix}; \; M_T(\lambda) = \begin{pmatrix} t_{11T}(\lambda) & t_{12T}(\lambda) \\ t_{21T}(\lambda) & t_{22T}(\lambda) \end{pmatrix} = M_1(\lambda) \cdot M_2(\lambda) \cdot \ldots \cdot M_n(\lambda) \tag{9}$$

The corresponding transfer functions for the reflection and transmission fields coefficients of the whole IBG are:

$$H_r(\lambda) = \frac{t_{21T}(\lambda)}{t_{11T}(\lambda)} = |H_r(\lambda)| e^{j\phi_r(\lambda)} = \sqrt{R(\lambda)} \cdot e^{j\phi_r(\lambda)} \tag{10}$$

$$H_t(\lambda) = \frac{1}{t_{11T}(\lambda)} = |H_t(\lambda)| e^{j\phi_t(\lambda)} = \sqrt{T(\lambda)} \cdot e^{j\phi_r(\lambda)} \tag{11}$$

where $R(\lambda)$ and $T(\lambda)$ represent reflectivity and transmissivity; and, $\phi_r(\lambda)$ and $\phi_t(\lambda)$ are the corresponding phase functions. From these IBG phase functions, the reflection and transmission group delay responses, GD, will be given by the following expressions:

$$\tau_r(\lambda) = \frac{\lambda^2}{2\pi c} \frac{d\phi_r(\lambda)}{d\lambda} \tag{12}$$

$$\tau_t(\lambda) = \frac{\lambda^2}{2\pi c} \frac{d\phi_t(\lambda)}{d\lambda} \tag{13}$$

Finally, as stated in [18], the ERI–TMM also provides the internal optical field distributions of the co-propagating and counter-propagating light waves through the IBG, offering two-dimensional plots of the reflected and transmitted average internal power versus the distance along the IBG axis and wavelength. These representations allow a deeper understanding of the microscopic behavior of these components.

## 3. Designing Chirped IBG

The conventional definition of chirp in Bragg Grating technology comes from the chirped FBG design. In this context, *chirp* is considered the variation in the grating period along the grating length to achieve higher bandwidths or to obtain spectra that exhibit a linear group delay [11]:

- Spectrum bandwidth broadening. Following the Bragg condition, a linear variation in $\Lambda_B$ will produce a linear variation on $\lambda_B$, and therefore an effect of increasing the total reflected bandwidth as the sum of the bandwidths for each $\lambda_B$.
- Linear group delay (GD). Due to the fact that different Bragg wavelengths are reflected at different locations of the grating, $\lambda_B(z)$, the time that every wavelength spends in the grating before achieving the phase-matching condition and gets reflected, is different and will determine the linear GD.

A more general definition of the chirp function in IBG technology would be the variation of the coupled Bragg wavelength along the grating according to Equation (1), where the variation in $\lambda_B(z)$ can be achieved by two alternatives: (a) varying the Bragg period in a similar way as in FBG; or (b) modifying the average effective refractive index along the length of the grating. Each strategy can be implemented through the modification of the suitable grating geometrical parameter.

Next, the procedure that allows the engineering of the chirped IBG parameters is presented. From here on, let us consider the design of a chirped IBG with a total bandwidth (BW: full width at 10% of maximum) of 20 nm, a GD of 20 ps, reflectivity around 90% and 1550 nm as the central wavelength. The parameters used to define the IBG are to be rectangular corrugation shape with maximum corrugation width of 10 nm and hyperbolic tangent (*tanh*) as the apodization function. $L$, $W_0$, and $\Lambda_B(z)$ will be determined from the specified BW and GD. Given the specifications of BW and GD for the chirped IBG, this procedure to design it will consist of:

- Using Equation (1) to calculate the Bragg grating period for the wavelengths of the spectral interval around 1550 nm depending on the intended BW, or the $n_{eff}$ needed for a particular central wavelength.
- Using the graphical representations (Figures 2 and 3) of $n_{eff}(\lambda,W)$ to determine the necessary width for a proposed $n_{eff}$, or the $n_{eff}$ required for a given $W$, at the wavelengths of interest. As an alternative, an analytical calculation can be made by making a polynomial fit of the matrix $N$ that defines $n_{eff}(\lambda,W)$; however, for the sake of illustration, the graphical method is presented here.
- The length $L$ of the IBG is estimated by evaluating the time it takes for the pulse to be reflected by the grating. This requires the use of the *group index* concept, which can be defined as the refractive index of the envelope of the optical pulse that propagates along the waveguide.

- Finally, ERI–TMM is applied to obtain the simulation results for reflectivity and group delay of the IBG.
- After checking the results, a fine adjustment can be made to the initial parameters in order to achieve a better transfer function, if necessary.

### 3.1. Chirp via Bragg Period Variation

The first design method consists of linearly varying the Bragg grating period along the grating length (maintaining a constant waveguide width, $W_0$ = 500 nm), which will cause a variation in $\lambda_B$ [9,10,19,20]. This can be expressed as:

$$\frac{d\lambda_B}{dz} = 2n_{eff}\frac{d\Lambda_B}{dz} \tag{14}$$

To obtain a total bandwidth of BW = 20 nm centered at 1550 nm, the initial $\lambda_B$ has to be $\lambda_{Bi}$ = 1540 nm and the final $\lambda_{Bf}$ = 1560 nm. Therefore, from the data contained in the matrix $N$, the corresponding initial and final $n_{eff}(\lambda,W)$ are: $n_{eff}$(1540 nm, 500 nm) = 2.4638 and $n_{eff}$(1560 nm, 500 nm) = 2.4412. These values have been highlighted in Figure 2 for $W$ = 500 nm (light blue line) with points A and B. The analogous concept is represented in Figure 3 by the line connecting points A and B. Thus, applying Equation (1), the Bragg period should be $\Lambda_{Bi}$ = 312.5 nm at the beginning, increasing linearly to a final value of $\Lambda_{Bf}$ = 319.5 nm.

The length of the device can be estimated by understanding that the GD is equal to the round-trip propagation time (because IBGs are devices that work in reflection, so the optical beam makes the round trip through the IBG):

$$t_{prop} = 2L\frac{n_g(1550, 500\text{ nm})}{c} \tag{15}$$

$$n_g(\lambda) = n_{eff}(\lambda) - \lambda\frac{dn_{eff}(\lambda)}{d\lambda} \tag{16}$$

where $c$ is the speed of light in a vacuum, and $n_g$ is the group index that must be calculated at a center wavelength of 1550 nm for a waveguide width of 500 nm. By means of Equation (16), the value obtained, after making the derivative of matrix $N'$ with respect to $\lambda$, is $n_g$ = 4.208. Thus, for a GD = 20 ps, $L$ is approximately equal to 2258 Bragg periods, resulting in the exact value $L$ = 712.44 µm (in the IBG literature, it is common to express length in Bragg periods). In chirped IBG designed by varying the Bragg period, this way of quantifying the grating period is, in fact, an approximation because of its own variation. Thus, the average Bragg period taken is the value for $\lambda_B$ = 1550 nm, i.e., $\Lambda_B$ = 316 nm.

Using these parameters and *tanh* as the apodization function, Figure 5a illustrates the design of the apodized chirped IBG, where the average waveguide width is kept constant and the grating period varies linearly along the length of the device. The proposed design is modeled by the ERI–TMM, and the results of the simulation are shown in Figure 5b. It can be noted that both BW and GD agree quite well with the objective design. Reflectivity is higher than 90% due to the long length of the IBG, and GD is quite linear because the ripples that could degrade the signal are significantly reduced by the apodization function [7,16,18].

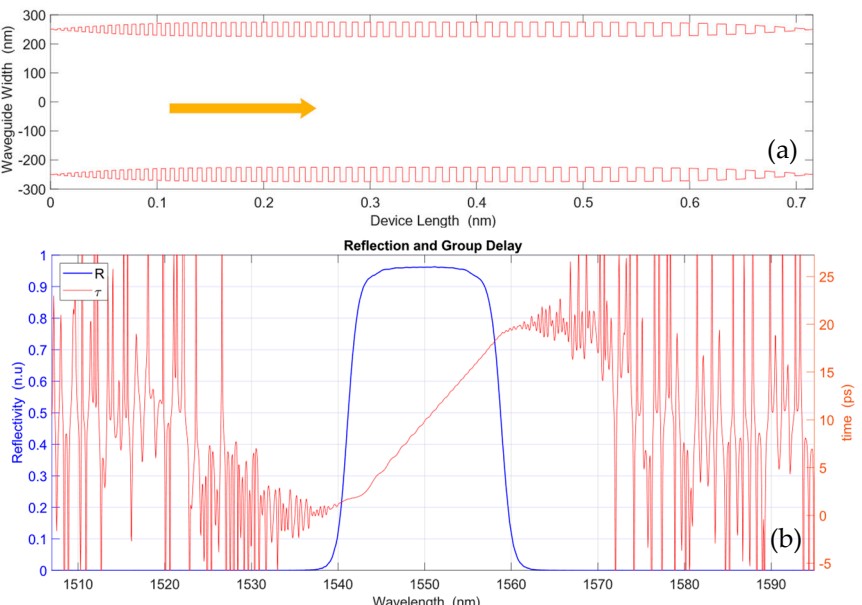

**Figure 5.** (**a**) Conceptual diagram of the apodized chirped IBG designed via Bragg grating period variation: initial Bragg period $\Lambda_{Bi}$ = 312.5 nm and final $\Lambda_{Bf}$ = 319.5 nm, $L = 2258\Lambda_B$, $\Delta W$ = 10 nm, $W_0$ = 500 nm, with *tanh* as the apodization function (grating period and corrugation width sizes have been enlarged for the sake of illustration). (**b**) Simulated spectrum of reflectivity and GD.

### 3.2. Chirp via IBG Waveguide Width Variation

According to the expression (1) Bragg wavelength can also be modified by changing the average effective refractive index along the grating length, $n_{eff}(z)$. As already explained, IBG technology can easily provide this $n_{eff}(z)$ through the waveguide width variation along the device length [21].

$$\frac{d\lambda_B}{dz} = 2\Lambda_B \frac{dn_{eff}}{dz} \tag{17}$$

Hence, the strategy in this case to obtain the chirp function is to increase (or decrease) the width of the waveguide by modifying $n_{eff}(z)$ and thus $\lambda_B(z)$, while keeping constant the Bragg grating period $\Lambda_B$. Thus, for a BW = 20 nm centered at 1550 nm and $\Lambda_B$ = 316 nm, it is necessary to select the appropriate waveguide width at $\lambda_{Bi}$ = 1540 nm and $\lambda_{Bf}$ = 1560 nm. In this case, expression (1) must be used to calculate the effective refractive index for each $\lambda_B$. In that way, results $n_{eff}$(1540 nm, 500 nm) = 2.4367 and $n_{eff}$(1560 nm, 500 nm) = 2.4684. Then, one must use Figure 3 to search for the necessary waveguide width at 1540 nm and 1560 nm (yellow and brown lines, points C and D) to achieve the required $n_{eff}$. These points are found to be $W_{0i}$ = 465 nm and $W_{0f}$ = 546 nm. The analogous concept is represented in Figure 2 by means of the line that connects points C and D. The device length will be computed taking the same $n_g$ = 4.208 (at wavelength 1550 nnm and 500 nm waveguide width) as in the previous case. Although it must be taken into account that, according to (17), the variation along the structure of the $n_{eff}$ would cause a $n_g(z)$; in light of this approximation, which does not affect the validity of the results, the value obtained again is $L$ = 2258 Bragg periods.

Figure 6a depicts a representation of the design of the new apodized chirped IBG, where the grating period is constant and the average waveguide width varies linearly along the length of the device. This design was introduced in the ERI-TMM for the chirped IBG modeling, being the results shown in Figure 6b. It can be observed that reflectivity and GD are very similar to the transfer function provided by the previous method and that, therefore, they match the proposed design.

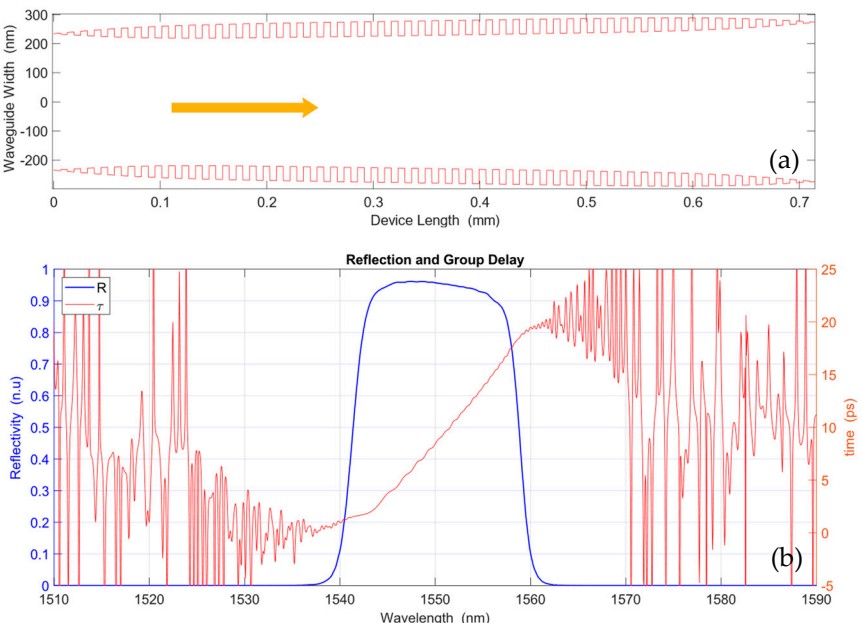

**Figure 6.** (**a**) Conceptual diagram of the apodized chirped IBG designed via waveguide width variation; initial width $W_{0i}$ = 435 nm and final $W_{0f}$ = 546 nm, $L = 2258\Lambda_B$, $\Delta W$ = 10 nm, constant $\Lambda_B$ = 316 nm and *tanh* as apodization function (grating period and corrugation width sizes have been enlarged for the sake of illustration). (**b**) Simulated spectrum of reflectivity and group delay (GD).

This method is less affected by manufacturing flaws, as will be explained in the following lines. It has been experimentally calculated that the standard variation in the dimensions of the manufactured features for SOI processes is 1.316 nm [22]. If this variation occurs in the *z*-axis, changes will appear in the Bragg period, which according to (1) would cause variations in the $\lambda_B$ in the interval [1543.5 nm, 1556.4 nm] for $\lambda_B$ = 1550 nm. If this variation occurs on the *x*-axis, there will be modifications in $n_{eff}$ due to its dependence on the IBG width. For a perturbation width of 10 nm, a tolerance of $\pm 1.316$ nm will cause variations in the $n_{eff}$ (considering a $n_{eff}$ = 2.4525) in the interval [2.4534, 2.4515], which will produce deviations in $\lambda_B$ in the range [1549.4 nm, 1550.5 nm]. That is, variations in the *z*-dimension affect the bandwidth much more than those that occur in the *x*-dimension. This is why performing the chirp by varying the average width of the waveguide (*x*-dimension) would allow a better control of the desired spectral response.

### 3.3. Chirp via IBG Bragg Grating Period and Waveguide Width Variation

Based on the two classical methods described above, we propose the combination of both to achieve similar results but keeping the effective refractive index constant for every Bragg wavelength along the grating length. Thereby, the waveguide will be modulated at the same time, increasing (or decreasing) the Bragg grating period and increasing (or decreasing) the average waveguide width. In this case, the parameter to keep constant is the average effective refractive index, $n_{eff}$. First, the Bragg grating period for $\lambda_{Bi}$ = 1540 nm and $\lambda_{Bf}$ = 1560 nm must be calculated, maintaining constant the effective refractive index along the grating length to $n_{eff}$(1550 nm, 500 nm) = 2.4525. Using Equation (1), the obtained value is $\Lambda_{Bi}$ = 314 nm at the beginning of the IBG, which will be increased linearly to a final Bragg grating period of $\Lambda_{Bf}$ = 318 nm. Next, the waveguide width at each wavelength can be obtained through Figure 3 (or Figure 2), points E and F. For a value of $n_{eff}$ = 2.4525, the waveguide widths must be $W_{0i}$ = 484.5 nm and $W_{0f}$ = 517.3 nm at the beginning and at the end of the chirped IBG. The value of $n_g$ will again be 4.208 for the same reasons as above, and $L$ will be approximated as in previous cases as 2258 Bragg periods.

In Figure 7, the simulation results for the new tanh-apodized chirped IBG design are shown. Figure 7a illustrates the average internal power distribution for the reflected

optical fields along the length of the device evaluated at every wavelength [18]. The red area in this two-dimensional plot indicates the location inside the chirped IBG where the different wavelengths are coupled from the co-propagating optical field to the counter-propagating optical field thanks to the phase-matching condition. It can be noted how shorter wavelengths are reflected at the beginning of the IBG and longer wavelengths at the end; hence the group delay increases with the wavelength, as shown in Figure 7b, which depicts the simulation result for the reflection transfer function, i.e., reflectivity as an amplitude response, and GD as a phase response.

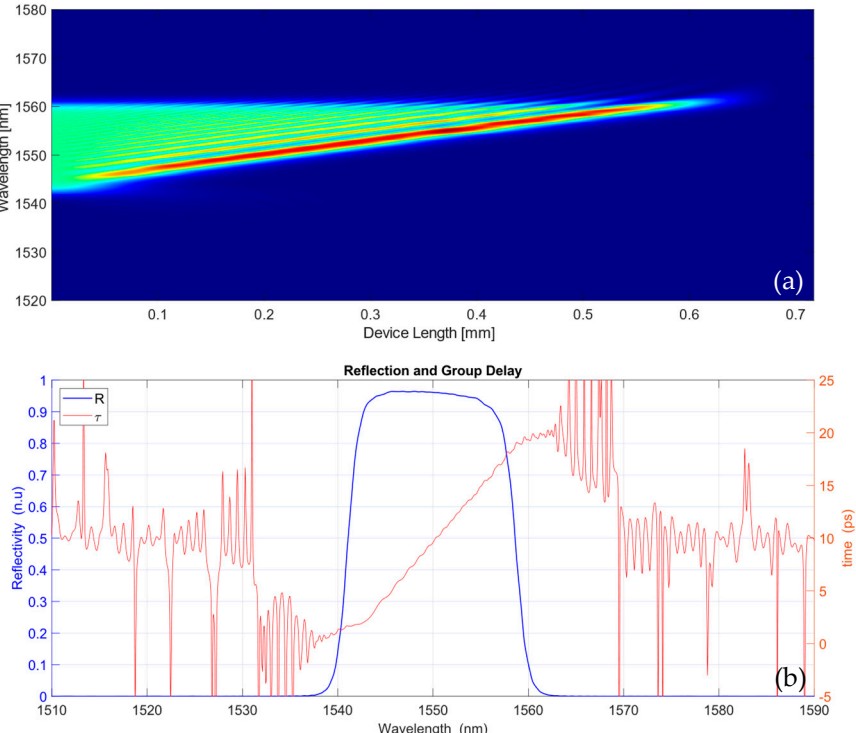

**Figure 7.** Simulation results for the new apodized chirped IBG design via Bragg grating period and waveguide width variation. IBG with initial $W_{0i}$ = 484.5 nm and $\Lambda_{Bi}$ = 314 nm, and final $W_{0f}$ = 517.3 nm and $\Lambda_{Bf}$ = 318 nm, $L = 2258\Lambda_B$, $\Delta W$ = 10 nm, constant $n_{eff}$ = 2.4525, $\Lambda_B$ = 316 nm, and *tanh* as the apodization function: (**a**) average internal power distribution for the reflected optical fields along the device length evaluated at each wavelength; and (**b**) spectrum of reflectivity and GD.

These results are equivalent to the ones obtained by previous methods, but with a minor modification of the geometric parameters of the IBG, both in the IBG average waveguide width and in the Bragg grating period. This fact has two main advantages: on the one hand, fewer manufacturing errors are likely to be made in the fabrication process, based on the explanation given above; and on the other hand, a higher bandwidth can be obtained.

## 4. Discussion

The results obtained demonstrate that the design process based on the study of the behavior of the effective refractive index and its dependence on the wavelength and width of the IBG (which governs the behavior of the light) is a feasible and novel way of tackling the problem of the geometrical parameters of a chirped IBG. In the literature on this subject, a chirped IBG is usually presented with some parameters already defined [10,11,19,20]; after that, a simulation is performed and the device is then fabricated to verify that the results are correct. In some studies, such as [9,21], this simulation was not even performed (or the authors did not present it); in these works, a device is manufactured and then characterized in order to assess, a posteriori, if the experimental results are suitable for a

specific application. This paper is intended to cover this initial part of the design, providing designers with the physical dimensions of the IBG before proceeding to the simulation and manufacturing process.

As a consequence of the two existing chirped IBG design methods and by taking advantage of the potential of the geometry modulation of IBG, we have devised a third method which consists of the modulation of the $n_{eff}$, both varying the Bragg period and the waveguide width linearly along the device length. The inclusion of the third chirp method is also innovative. Varying the Bragg period combined with varying the IBG width provides the same results as those obtained with the two standard methods separately [9,21]. In addition, it has the advantage that varying the geometry of the waveguide in two dimensions ($x$-dimension and $z$-dimension) requires less variation in each of them (as demonstrated in Section 3.3). As indicated in [23], phase noise caused by unwanted grating phase variations (Bragg period variation) can distort the IBG spectrum. Therefore, smaller variation in the Bragg period (owing to simultaneous variation in the waveguide width) should decrease the probability of error. This has been verified theoretically in Section 3.3, where it has been shown that errors on the $x$-axis (variation in the width of the waveguide) affect the signal less than errors on the $z$-axis (variation in the Bragg period).

Another advantage of this new method of variation in two dimensions is that it allows wider bandwidths to be achieved, thanks to the addition of two effects: the Bragg period, which varies the Bragg wavelength, and the effective refractive index variation, which also causes a Bragg wavelength variation.

This work could be included in a fundamental study of the IBG. Future work could be directed toward the fabrication of a chirped IBG using this method. Meanwhile, a Lumerical 3D simulation could also be accomplished to verify the theoretical results obtained in this work.

## 5. Conclusions

ERI–TMM has proven to be the simulation method that is best suited to the characteristics of IBGs [24,25], given its ability to accurately map the geometric structure of the IBG onto the effective refractive index profile of the grating. To do this, it is necessary to first characterize the waveguide using a tool such as Lumerical®, which allows us to obtain an effective refractive index that will depend on the wavelength and width of the waveguide. With this method, we have reviewed and analyzed the two traditional methods for chirping an IBG.

We have proposed a process that allows us to determine the IBG width and length parameters, as well as the Bragg period, prior to the simulation process; then, by applying the ERI-TMM, we have calculated the related reflectivity and group delay. Simulation results allow us to adjust, if necessary, the width and length of the IBG to fine-tune the transfer function. Finally, we have also devised an alternative method to design chirped IBG that keeps constant the $n_{eff}$ for every Bragg wavelength, while at the same time varying the Bragg period and the waveguide width.

**Author Contributions:** Conceptualization, J.Á.P. and A.C.; methodology, J.Á.P. and A.C.; software, J.Á.P.; validation, J.Á.P. and A.C.; formal analysis, J.Á.P. and A.C.; investigation, J.Á.P. and A.C.; resources, A.C.; writing—original draft preparation, J.Á.P.; writing—review and editing, J.Á.P. and A.C.; visualization J.Á.P. and A.C.; project administration, A.C. All authors have read and agreed to the published version of the manuscript.

**Funding:** This research received no external funding.

**Institutional Review Board Statement:** Not applicable.

**Informed Consent Statement:** Not applicable.

**Data Availability Statement:** The original contributions presented in the study are included in the article, further inquiries can be directed to the corresponding author.

**Conflicts of Interest:** The authors declare no conflicts of interest.

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
