# Peer review of "Chirped Integrated Bragg Grating Design"

_photonics, doi:10.3390/photonics11050476_

Round 1
Reviewer 1 Report
Comments and Suggestions for Authors
The paper presented detailed insights into the Transfer Matrix Method (TMM) for designing integrated chirped gratings on a SOI chip. The systematic approach outlined in the paper, along with three examples of chirped grating designs, was thorough. TMM is an effective method in modeling long IBG structure as it is very computationally efficient in comparison with FDTD method. Although integrated Bragg grating (IBG) designs using TMM have been extensively studied, the paper provides a viable approach that can be followed and implemented by those less familiar with this method. Detailed comments are provided below:
- The paper's English is understandable, but there is room for improvement in both technical clarity and grammar proficiency.
- Matrix (N) was introduced in line 105 on page 3, but its explanation was not provided until lines 127 to 130.
- Figure 3 plots Neff as a function of waveguide width in the range of 400nm to 600nm. The step of waveguide width change needs clarification, and if discrete widths were chosen for Lumerical Neff calculation, those data points should be clearly marked on the plot.
- Equation 4 introduces the M(lambda) matrix, where t11i, t12i, t21i, and t22i are the four matrix components. These t*** values are not defined, and the subscript needs to be explained.
- Hr and Ht in Equations 5 and 6 need to be defined. The physical significance of Hr and Ht should be clarified.
- Equation 11 relates the group index ng with n(lambda). What is n(lambda)? Is it the material index or should it be Neff(lambda)?
- In line 213, the author states that ng was obtained by taking the derivative of Neff with respect to lambda, while Equation (11) states ng calculation differently. This discrepancy needs to be addressed.
- In section 3.1, line 215 mentions obtaining ng = 4.208 for chirp design via varying period (method 1). What is the ng obtained in chirping by varying waveguide width (section 3.2) and the combination of waveguide width and period chirping (section 3.3)?
- In section 3.1, line 215 states "Thus, for a GD = 20ps, L is approximately equal to 2258 Bragg periods." By theory, L would vary using the other two chirping strategies due to differences in dispersion, but the paper reports identical L = 2258 periods. Would the authors provide further explanation on this?
- The comparison among the three chirping methods in the last paragraph (lines 306–312) should be included in section 3. What is the benefit of varying the combination of period and waveguide width?
- The conclusion section needs improvement.
- Point A, B, C, D, E, and F are marked on the graph of Fig. 2 and 3, but only A and B are discussed in the text. An illustration of the remaining data points should be provided.
- IBG is also used in active photonic devices such as slow-light EO modulators. I recommend adding these references to the introduction section: Stephen R. Anderson, IEEE Photonics Journal, Vol. 14, No. 4, August 2022, and O. Jafari, IEEE Photonics Technology Letters, Vol. 32, No. 8, April 15, 2020. TMM has also been used in sub-wavelength grating design. I recommend adding the reference: Hao Sun et al., Journal of Lightwave Technology, Vol. 40, No. 20, October 15, 2022.
- Ideally, the reported TMM in IBG design should be validated by fabricated devices. In the absence of experimental vs theory comparison, I would recommend Lumerical 2.5D simulations (3D if computing is possible) of the proposed structures to serve as theoretical modeling verification.
- The paper's English is understandable, but there is room for improvement in both technical clarity and grammar proficiency.
Reviewer 2 Report
Comments and Suggestions for Authors
Extremely clearly written manuscript - both in the language and scientific soundness. Unfortunately it is nowadays a very rare case that authors can tell a coherent story where every sentence is meaningfull and has a logical connection to the precious one and to the next one. J. A. Praena and A. Carballar can do it exceptionally well or they can use effectively the modern AI language tools. Great job!
Here are two issues that could be explained more elaborately and some minor suggestions to fine-tune the manuscript:
- can you explain more elaborately the advantage and the use-case of the third method? For what application the constant effective index along the grating would be usefull? Second, if I understand correctly, the third method has no advantage for precise manufacturing since device dimensions in both directions: x and z are varied. The latter leads to broadening of the spectrum.
- Eq. 11: is it n or n_eff on the right hand side ? If this is n_eff and the derivative is zero, then wouldn't it influence the group delay? Is it a distinctive feature of the third design with n_eff=const? Can you add a comment on it in the last chapter (3.3) or in conclusion?
- line 25: do you mean "high polarization extinction ratio"?
- Caption of the FIgure 1b: you don't mean "mode distribution" but probably "energy density distribution of the quasi-TE mode".
- Fig. 3 and 3 - it would look nicer if the gaph has full frame (add frame lines up and right)
- Caption of Fig. 2 and 3: since you mentioned once in the main text that the numerical simulation was done with Lumerical software, don't write it here again. It is better if you write "with numerical simulation" without advertising the company name
- Eq. 7 and 8: c instead of c^2 ? I don't know the equations but just the units of tau are not correct.
- line 202, 203 and further in the main text: add space between the two numbers in parenthesis
- line 212 - delete the tab in the beginning of the line
- line 255: standard variation of what? Maybe: standard variation of the dimentions of manufactured features ?
Comments on the Quality of English LanguageSome minor language issues detected:
- line 60: do you mean height (instead of high)? In the same sentence there is "waveguide width" thus "height" would be correct.
- line 81: is composed of
- line 92: dependence on
- line 122: effective refractive index neff(z) along the device length
- line 125: first instead of firstly
- line 129: dependence on
- line 212 speed of light in vacuum ("the" is surplus)
- line 279: move "are shown" to the end of the sentence
- line 299: "function" is surplus
Round 2
Reviewer 1 Report
Comments and Suggestions for Authors
The revised draft has addressed all my previous questions. In figure 1 the authors add a plot with caption of "Energy density distribution of the quasi-TE mode inside the strip waveguide (linear scale). " Will the author show more clearly on the units of the color bar as it seems normalized to 1.
Comments on the Quality of English LanguageThe English of the draft is overall acceptable, but can be further polished.
Author Response
Thank you for your suggestions.
a) We have revised the use of English and grammar in the document with the help of colleagues with a great knowledge of English. We hope it is now more correct.
b) Regarding the comment on Figure 1, we have indicated in the title of the figure that the energy density distribution is normalized.